

Earth System
Dynamics

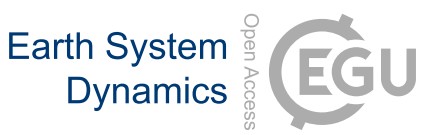

# Minimal dynamical systems model of the Northern Hemisphere jet stream via embedding of climate data

**Davide Faranda[1,2], Yuzuru Sato[3,2], Gabriele Messori[4,5], Nicholas R. Moloney[6,2], and Pascal Yiou[1]**

[1]Laboratoire des Sciences du Climat et de l'Environnement, UMR 8212 CEA-CNRS-UVSQ,
Université Paris-Saclay, IPSL, 91191 Gif-sur-Yvette, France
[2]London Mathematical Laboratory, 8 Margravine Gardens, London, W6 8RH, UK
[3]RIES/Department of Mathematics, Hokkaido University, Kita 20 Nichi 10, Kita-ku, Sapporo 001-0020, Japan
[4]Department of Earth Sciences, Uppsala University, 752 33 Uppsala, Sweden
[5]Department of Meteorology and Bolin Centre for Climate Research,
Stockholm University, Stockholm, Sweden
[6]Department of Mathematics and Statistics, University of Reading, Reading, RG6 6AX, UK

**Correspondence:** Davide Faranda (davide.faranda@lsce.ipsl.fr)

**Abstract.** TS1 CE1 We derive a minimal dynamical systems model for the Northern Hemisphere midlatitude jet dynamics by embedding atmospheric data and investigate its properties (bifurcation structure, stability, local dimensions) for different atmospheric flow regimes. The derivation is a three-step process: first, we obtain a 1-D description of the midlatitude jet stream by computing the position of the jet at each longitude using ERA-Interim. Next, we use the embedding procedure to derive a map of the local jet position dynamics. Finally, we introduce the coupling and stochastic effects deriving from both atmospheric turbulence and topographic disturbances to the jet. We then analyze the dynamical properties of the model in different regimes: one that gives the closest representation of the properties extracted from real data; one featuring a stronger jet (strong coupling); one featuring a weaker jet (weak coupling); and one with modified topography. Our model, notwithstanding its simplicity, provides an instructive description of the dynamical properties of the atmospheric jet.

## 1 Introduction

Jet streams are narrow, fast-flowing westerly air currents near the tropopause. They are a major feature of the large-scale atmospheric circulation and modulate the frequency, severity and persistence of weather events across the extratropics (e.g., Röthlisberger et al., 2016). Their location and intensity also affects commercial aviation and shipping (Reiter and Nania, 1964; Hadlock and Kreitzberg, 1988; Williams and Joshi, 2013). Two types of atmospheric jets can be identified: thermally driven subtropical jets, and eddy-driven jets associated with baroclinic instability at the polar front. In the Northern Hemisphere (NH), the two are not always clearly separated (Lee and Kim, 2003), and when considering monthly or longer time averages, a single, spiral-shaped jet structure emerges (e.g., Archer and Caldeira, 2008). In this paper we consider a single NH jet (NHJ), rather than attempting to separate the subtropical and eddy-driven jets (e.g., Belmecheri et al., 2017).

Even though the climatological NHJ is a westerly flow, it can present large meanders on synoptic timescales (e.g., Koch et al., 2006; Röthlisberger et al., 2016). These can cause the local flow to become predominantly meridional or can even determine a splitting or breaking of the jet (Haines and Malanotte-Rizzoli, 1991). The occurrence of these large meanders in the jet is often associated with events such as temperature and precipitation extremes (e.g., Dole et al., 2011; Screen and Simmonds, 2014). Although jet dynamics are well understood in a climatological sense, our insights into dynamical features such as jet splitting or meandering are still limited.

The dynamics of meanders and split jets has often been framed in terms of transitions between zonal and blocked flows since the seminal work by Charney and DeVore (1979). Legras and Ghil (1985) and Ghil (1987) used an intermediate complexity barotropic model with dissipation forcing and topography and observed two distinct equilibria associated with the zonal and blocked flows. Similar mechanisms have been proposed by Mo and Ghil (1988) and then performed in experimental facilities (Weeks et al., 2000). However, there is no consensus about the nature of flow multistability, and a wide range of theoretical explanations and models have been proposed (e.g., Tung and Lindzen, 1979; Simmons et al., 1983; Frederiksen, 1982; Faranda et al., 2016b). Moreover, jet dynamics have been described as a manifestation of multiple equilibria in asymmetrically forced flows (Hansen, 1986) or as a result of soliton–modon structures (McWilliams et al., 1981).

In order to advance our understanding of the jet dynamics, we employ a low-dimensional dynamical systems model derived from reanalysis data. The best-known example of a low-dimensional model for atmospheric phenomena is Lorenz' simple three-dimensional system representing some features of Rayleigh–Bénard convection (Lorenz, 1963). Thereafter, simple dynamical systems models have been devised to study El Niño (Penland and Matrosova, 1994), ocean–atmosphere interactions (Dijkstra and Ghil, 2005), climate tipping points (Stommel, 1961; Benzi et al., 1982), large-scale atmospheric motions (Lorenz, 1984, 1996) and many other phenomena. The goal of these investigations was not to provide the most realistic representation of the relevant systems but rather to capture key emerging behaviors (such as chaos, intermittency, multistability). The main drawback of those investigations was the weakness of the connection between models and real data due to the scarcity of observations as well as theoretical limitations. Until very recently, there was a strong case against the use of embedding techniques to derive low-dimensional models from experimental data (Letellier et al., 2006). This opposition was motivated by a long sequence of papers that appeared between 1984 and 1991. The initial claim that low-dimensional models for complex phenomena could be derived using a very small numbers of variables (see e.g., Nicolis and Nicolis, 1984; Fraedrich, 1986) was disproved by rigorous numerical computations by Grassberger (1986) and Lorenz (1991).

Progress in data quality and availability and the advent of stochastic dynamical systems have renewed the attention for data embedding. Recently, Faranda et al. (2017c) have shown that embedding techniques can yield effective low-dimensional dynamics provided that the chosen observables reflect the symmetries of the system and that small-scale (sub-grid) dynamics are represented as stochastic perturbations. Here, we use these results to develop a minimal model of the effective dynamics of the midlatitude jet. This is useful to explore a range of possible behaviors beyond those displayed in the available data that could have appeared in past climates and could appear again in future climates. In analogy to the model derived by Faranda et al. (2017c) for the von Karman turbulent flow, the jet model is based on a coupled map lattice (CML; see Appendix A). Each element of the lattice reflects the dynamics of the jet at each longitude. Such a model does not require physical sub-grid terms a priori but only if they are found to be essential to capture the large-scale phenomenology – which we show is not the case. We then evaluate how this model represents key dynamical features of the jet, namely its stability, the statistics of splitting or breaking and the response to topographical features, and we relate the results back to the original ERA-Interim data.

First, we provide the details of the ERA-Interim data and of the jet detection algorithm (Sect. 2). We then present the stochastic coupled lattice map model and compute its bifurcation structure (Sect. 3). Next, we introduce some instantaneous dynamical indicators (Sect. 4) and use them to relate the conceptual model to more complex climate models and reanalysis data (Sect. 5). Finally, we highlight the open questions our results can answer and the new questions they pose (Sect. 6).

## 2   Data and methods

### 2.1   ERA-Interim data and jet position algorithm

The analysis is based on the European Centre for Medium Range Weather Forecasts's ERA-Interim (Dee et al., 2011). We consider daily data with a 1° horizontal resolution over the period 1979–2016.

The jet position is diagnosed through a modified version of the approach by Woollings et al. (2010). We take daily mean wind speed averaged over 200–400 hPa and apply a 10 d low-pass Lanczos filter (Duchon, 1979). We then identify the latitudinal position of the jet at every longitude as the location of the strongest wind over the band 15–75° N. This approach is intended to provide a "raw" measure of the jet variability. We then consider the longitude and time dependence of the latitude of the jet to monitor its waviness.

We define an index of large jet meanders, or breaks (breaking index, BRI), as the daily number of meridional variations in jet position of more than 10° of latitude across adjacent longitude grid points, except at longitude 0. The analysis has been repeated for BRI thresholds between 5 and 15°, with no significant qualitative differences.

Figure 1 shows a snapshot of the jet position on 4 February 1979, together with the time series of the daily jet position recorded in 1979 at longitude 120° W. An animation of the jet location for the year 1980 is provided as a supplementary video. Both the time series and the snapshot show large jumps in the jet position. A qualitative analysis of the jet position data suggests that the jet fluctuates around a central latitude (Central Jet, CJ) and seldom shifts to more northerly (NJ) or southerly (SJ) latitudes.

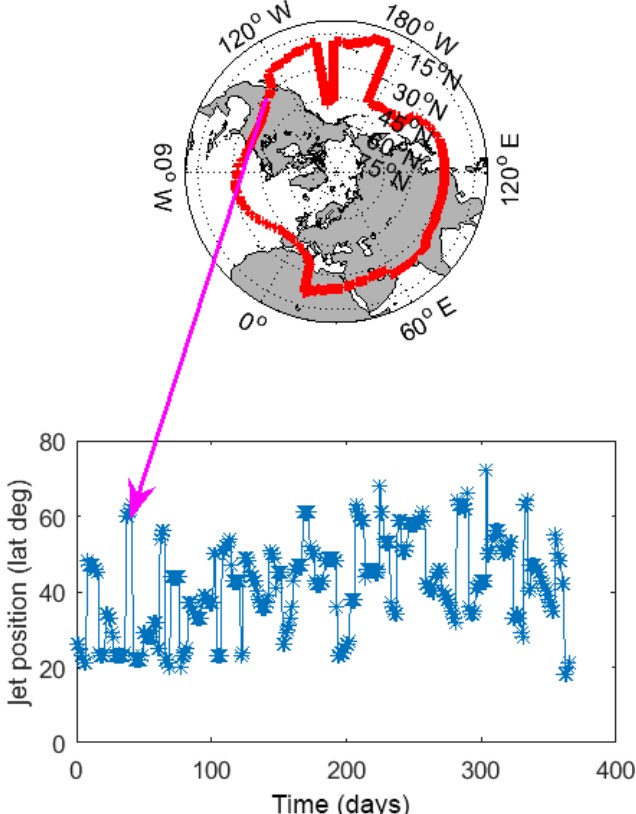

**Figure 1.** Snapshot of the jet position extracted from the ERA-Interim data set on 4 February 1979 and time series of the jet position for the year 1979, recorded at longitude 120° W.

In order to embed the data and derive the effective maps of the dynamics, we remove the seasonal cycle from the data by subtracting, longitude by longitude, the average meridional position for each calendar day and dividing by the standard deviation. For the deseasonalized data, the dimensionless threshold for the computation of BRI corresponding to about 10° latitude is $|x| > 1$.

## 2.2 Local dynamical systems metrics

Our analysis leverages two recently developed dynamical systems metrics, namely the local dimension of the attractor $d$ and the stability or persistence of phase-space trajectories $\theta^{-1}$. Instantaneity in time corresponds to locality in phase space, such that a value of $d$ and $\theta^{-1}$ can be computed for a given variable (in our case the jet position data) at every time step. $d$ is a proxy for the system's active number of degrees of freedom. It provides information on how the system can reach a given state and how it can evolve from such state. $\theta^{-1}$ describes the persistence of a state in time, thus providing complementary information to $d$.

### 2.2.1 Local dimension

The local dimension is estimated by making use of extreme value statistics applied to Poincaré recurrences. The Freitas et al. (2010) theorem, modified by Lucarini et al. (2012), states that the probability of entering a hyperball with a small radius centered on a state $\zeta$ on a chaotic attractor obeys a generalized Pareto distribution (Pickands III, 1975). In order to compute this probability empirically, we first calculate the series of distances dist($x(t),\zeta$) between the point on the attractor $\zeta$ and all other points $x(t)$ on the trajectory. This series is transformed via the distance function:

$$g(x(t)) = -\log(\text{dist}(x(t), \zeta)), \tag{1}$$

such that close recurrences of $\zeta$ correspond to large values of $g(x(t))$ (Collet and Eckmann, 2009). Thus, the probability of entering a small hyperball around $\zeta$ is transformed into the probability of exceeding a high threshold $s(q)$, where $q$ is a percentile of the series $g(x(t))$ itself. In the limit of an infinitely long trajectory, it can be shown that the choice $g(x(t))$ in Eq. (1) locks this probability into the exponential member of the generalized Pareto distribution:

$$\Pr(z > s(q)) \simeq \exp\left[-\vartheta(\zeta)\left(\frac{z - \mu(\zeta)}{\beta(\zeta)}\right)\right], \tag{2}$$

where $z = g(x(t))$ and $\mu$ and $\beta$ (obtained via fitting) depend on the point $\zeta$. These are the location and the scale parameters of the distribution. Remarkably, $\beta(\zeta) = 1/d(\zeta)$, where $d(\zeta)$ is the local dimension around the point $\zeta$. This result has been recently applied to a range of atmospheric and oceanic fields (e.g., Faranda et al., 2017b, a; Messori et al., 2017; Faranda et al., 2019a, b). In this paper, we use the quantile 0.975 of the series $g(x(t))$ to determine $q$. Our results are robust with respect to reasonable changes in this quantile.

### 2.2.2 Local persistence

Extreme value statistics also allow estimating the persistence of a given state $\zeta$, by inspecting the temporal evolution of the dynamics around it. A measure of persistence around $\zeta$ can be obtained from the mean residence time of the trajectory within the neighborhood of $\zeta$. To quantify this, we employ the so-called extremal index $\vartheta$ (Freitas et al., 2012; Faranda et al., 2016a): an adimensional parameter $0 < \vartheta(\zeta) < 1$ which can be interpreted as the inverse of the mean residence time. We can then compute $\theta^{-1}(\zeta) = dt/\vartheta(\zeta)$, where $dt$ is the time step of our data. Heuristically, if the $i$th visit to the neighborhood of $\zeta$ lasts $\tau_i$ consecutive time steps and $N$ such visits are made in total, then $\theta^{-1} \approx (1/N)\sum_i \tau_i$. In practice, instead of this naive estimator, we compute the extremal index using the likelihood estimator of Süveges (2007). $\theta = 0$ corresponds to a stable fixed point of the dynamics, so that the trajectory resides an infinite amount of time in the neighborhood of $\zeta$. $\theta = 1$ corresponds to a trajectory residing in the neighborhood of $\zeta$ for

only one time step per visit. The estimate of $\theta$ is thus sensitive to the d$t$ used. If d$t$ is too large, the time dependence structure is unresolved and $\theta$ will be close to 1. Conversely, if d$t$ is too small, $\theta$ will be close to 0. Faranda et al. (2017b) observed that $\theta$ varies between 0.3 and 0.5, when d$t = 1$ day, for sea-level pressure fields over the North Atlantic. In this work, we use the same d$t$.

## 3  Derivation of the lattice jet model

### 3.1  Model framework

While not directly issued from the Navier–Stokes equations, our framework builds on concrete physical hypotheses, namely that (i) the physics of the jet is the same at every longitude and is only slightly modified by the presence of topographical constraints, (ii) the jet can experience sudden breaks and shifts from its central position (CJ) to northern (NJ) or southern latitudes (SJ), (iii) the jet must propagate to the west, and (iv) smaller-scale phenomena, such as turbulence and baroclinic waves, will be introduced in the model only if necessary to reproduce the effective dynamics in the data. This latter point is fundamentally different from the philosophy of direct numerical simulations.

We construct our model starting from the local time series of the non-dimensionalized jet position $x$ measured at each longitude $i$ and time $t$. We use the simplest possible embedding procedure (see Appendix B), which consists of plotting the return map $x_t^{(i)}$ vs. $x_{t+1}^{(i)}$ (an example is shown in Fig. 2) and searching for a function $f^{(i)}$ such that $x_{t+1}^{(i)} = f^{(i)}(x_t^{(i)})$. The first thing to verify is that the same functional form $f^{(i)}$ may be used at all longitudes $i$. This is equivalent to asking that there is only one dynamic driving the jet independently of the location. With the choice

$$f^{(i)}(x) = \begin{cases} -\dfrac{A(A+x)}{A-c} + r^{(i)}, & x < -c, \\ \sinh(\beta x) + r^{(i)}, & -c \leq x \leq c, \\ \dfrac{A(A-x)}{A-c} + r^{(i)}, & c < x, \end{cases} \tag{3}$$

where we have dropped the dependencies of $x$ for clarity, the parameters can be fixed at all longitudes as $\beta = 0.75$, $A = 3$, and $c = \sinh^{-1}(A)/\beta \approx 2.4246$. Even though the functional form of $f^{(i)}$ is independent of longitude, a dependence on $i$ remains in the form of the parameter $r^{(i)}$, which represents the effects of topography in terms of spatial inhomogeneities of the local dynamics. As a first-order approximation, we consider only the difference between land and ocean and assign one of two discrete values to each $r^{(i)}$. The choice of the function $f^{(i)}$ is not unique; however, the one we propose here is a suitable option that satisfies hypotheses (i) and (ii) above. In order to reproduce the eastward propagation of the jet (hypothesis iii), we introduce the coupled map lattice (CML; see Kaneko, 1983, and Appendix A):

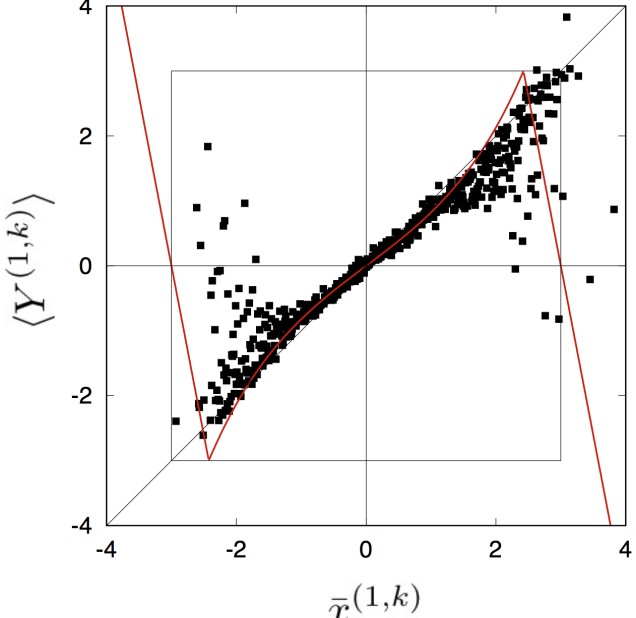

**Figure 2.** The average return map extracted from the data at longitude $i = 1$. This is constructed by coarse-graining the state space at $i = 1$ into $M$ partitions $L_k^{(1)}$ ($k = 1, 2, \ldots, M$). We then define $\overline{x}^{(1,k)}$ as the midpoint of the partition $L_k^{(1)}$, and $Y^{(1,k)} = \{x_t^{(1)} | x_{t-1}^{(1)} \in L_k^{(1)}\}$ ($t = 2, 3, \ldots$). The black dots represent $(\overline{x}^{(1,k)}, \langle Y^{(1,k)} \rangle)$ for $k = 1, 2, \ldots, 500$, where $\langle \cdot \rangle$ is the average over the elements of $Y^{(1,k)}$, computed based on the observed data. The red line represents the approximated average return map $\langle Y^{(1,k)} \rangle = f^{(1)}(\overline{x}^{(1,k)})$ when $|x| \leq c$. In the region $|x| > c$, we assume linear reflection effects. As a result, we have the return map $f^{(1)}$ in Eq. (3).

$$x_{t+1}^{(i)} = (1-\epsilon)f^{(i)}\left(x_t^{(i)}\right) + \epsilon f^{(i-1)}\left(x_t^{(i-1)}\right),$$
$$(i = 1, 2, \ldots, N;\ t = 1, 2, \ldots). \tag{4}$$

With this geometry, the dynamics are divided into $N = 360$ cells. Periodic boundary conditions are applied at $N = 360$. The dynamics in each cell $i$ are time-independent but perturbed by the cell $i-1$ (i.e., its neighbor to the west) with intensity $\epsilon$, which we estimate and scale based on the observed data. This further implies that our reference length-scale in the model corresponds to that of $1°$ longitude in the midlatitudes, namely of the order of 100 km.

### 3.2  Sub-grid feedbacks to jet dynamics

If we perform a numerical simulation of Eq. (4), the dynamics are fixed to one of the three states (CJ, SJ, NJ), depending on the value of $\epsilon$. This means that the role of small-scale perturbations in triggering the transitions between the states is fundamental. We therefore have to include an additive noise term $\xi_t^{(i)}$ in Eq. (4):

$$x_{t+1}^{(i)} = (1-\epsilon) f^{(i)}\left(x_t^{(i)}\right) + \epsilon f^{(i-1)}\left(x_t^{(i-1)}\right) + \xi_t^{(i)},$$
$$(i = 1, 2, \ldots, N; \ t = 1, 2, \ldots). \tag{5}$$

The noise is a fundamental ingredient for the breaking of the jet and the transition between zonal and blocked states, as shown in tank experiments and numerical simulations (Jacoby et al., 2011). Physically, noise arises from key sub-grid processes that affect the jet dynamics, such as convection or the interaction between the jet stream and gravity waves (Williams et al., 2003, 2005). Translated to our model with a reference spatial scale of the order of 100 km these phenomena, ranging from a few meters to a few kilometers, imply a perturbation in the range $10^{-4} < \nu < 10^{-3}$. Several sub-grid parametrization of turbulence exists: the seminal works of Kraichnan (1961) and Thomson (1987) showed that if large scales are represented by a deterministic term, a single random variable can drive the turbulence term. This means that Langevin model representations are appropriate to describe turbulent eddies (McComb, 1990; Frederiksen and Davies, 1997). Following these ideas, we model $\nu_t^{(i)} \in [-\delta, \delta]$ as a uniform random variable.

However, considering small-scale turbulent disturbances to the jet dynamics is not sufficient to reproduce the blocking and breaking of the jet. Even if the introduction of $\nu$ as a stochastic term can account for the direct Kolmogorov turbulent cascade (Kolmogorov, 1941), the jet dynamics are also driven by the effect of an inverse cascade transferring energy to large scales via baroclinic waves (Held and Larichev, 1996).

Baroclinic activity is associated with extratropical cyclones and anticyclones, on scales of the order of $10^3$ km. These can affect the jet position by several degrees of latitude. Again, there is no unique way to model baroclinic waves in our framework. We follow the rationale of multiscale parametrizations as they can be theoretically justified (e.g., Wouters and Lucarini, 2013; Kitsios and Frederiksen, 2019) and are numerically efficient (Faranda et al., 2014). The simple introduction of another source of noise $\eta^{(i)}$, acting at intermediate scales (i.e., between the scale of the jet and the scale of turbulence), is enough to obtain reliable jet breaking dynamics (see Sect. 4.1). The simplest choice for $\eta_t^{(i)} \in [-\mu, \mu]$ is a block noise taking the same value over bl CE2 cells (the one-dimensionalized size of cyclones or anticyclones; see Appendix B) and obeying the uniform distribution. Another choice for modeling baroclinic disturbances to the jet could be to introduce a second deterministic equation, weakly coupled with the jet position. However, this choice requires additional hypotheses and parameters and does not emerge naturally from the embedding procedure used to derive the dynamics of $x$.

The minimal sub-grid parametrization can thus be written in the following form:

$$\xi_t^{(i)} = \nu_t^{(i)} + \eta_t^{(i)}, \tag{6}$$

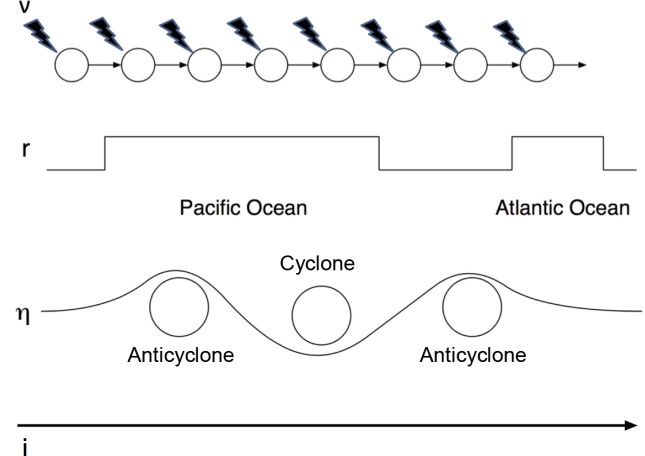

**Figure 3.** Schematic representation of noise contributions to the CML model (Eqs. 3 and 6): $\nu$ represents local turbulent disturbances, $r$ topographical features, $\eta$ baroclinic eddies and $i$ longitudinal positions.

where $\nu_t^{(i)}$ and $\eta_t^{(i)}$ model the effects of small turbulent disturbances and baroclinic eddies, respectively (Fig. 3). The noise terms are discussed further in Appendix B.

## 3.3 Local dynamics

Owing to the unidirectional coupling in our model and to the large $N$, the local dynamics can be approximated by a nonautonomous dynamical system:

$$x_{t+1}^{(i)} \simeq f^{(i)}\left(x_t^{(i)}\right) + p_t^{(i)}, \tag{7}$$

where a nonautonomous external force $p_t^{(i)}$ is given by

$$p_t^{(i)} = \epsilon \left[ f^{(i-1)}\left(x_t^{(i-1)}\right) - f^{(i)}\left(x_t^{(i)}\right) \right] + \nu_t^{(i)} + \eta_t^{(i)}. \tag{8}$$

When $|p_t^{(i)}| \to 0$, the local dynamics has a stable fixed point at $x \approx 0$ and two unstable chaotic sets near $x \approx \pm 2$. When $p_t^{(i)} > 0$, the resulting perturbed dynamics may exhibit escape behavior from the fixed point to the chaotic regions with positive Lyapunov exponents. Figure 4 shows the bifurcation diagrams as a function of $\kappa$ over land ($r^{(i)} = -0.02$) and ocean ($r^{(i)} = 0$; see Appendix B) obtained by approximating the external perturbation $p_t^{(i)}$ as a random variable obeying the uniform distribution in $[-\kappa, \kappa]$. They both indicate a bifurcation to chaotic and partially chaotic behavior (Sato et al.). The different value of $r^{(i)}$ over land gives rise to an asymmetry in the invariant sets, namely the sets delimiting the accessible region of the dynamics with respect to all possible external perturbations $p_t^{(i)}$. In Fig. 4, these dynamically reachable regions are depicted in gray, while a realization of the dynamics is depicted by the black dots. With $r^{(i)} = -0.02$ over land and $0.1574 < \kappa < 0.1985$, there is a small chance

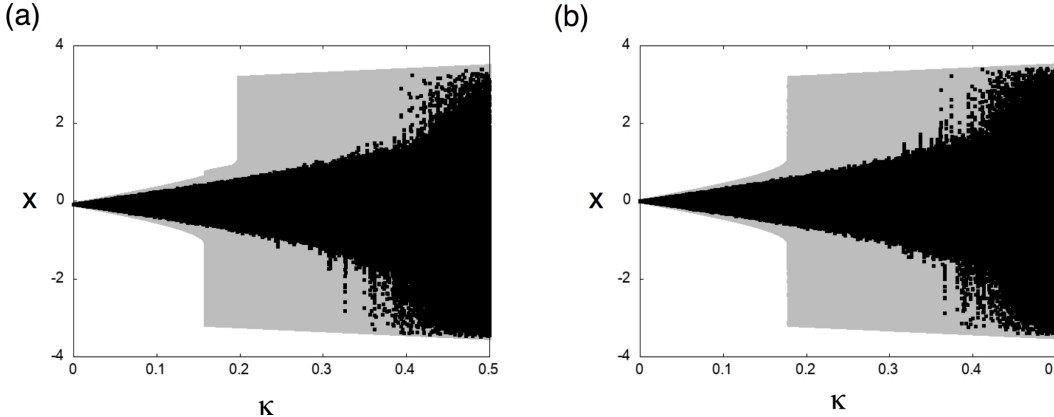

**Figure 4.** Bifurcation diagrams as a function of $\kappa$ for **(a)** land ($r^{(i)} = -0.02$) and **(b)** ocean ($r^{(i)} = 0.0$). The gray regions delimit the accessible region of the dynamics with respect to all possible external forcings. A realization of the dynamics is depicted by the black dots. For $r^{(i)} = -0.02$ and $0.1574 < \kappa < 0.1985$, there is a small chance of reaching SJ positions and no chance of reaching NJ positions.

of reaching SJ positions and no chance of reaching NJ positions. This is reflected in the skewed distribution of $x_t^{(i)}$. For the sake of conciseness, we do not report the detailed bifurcation analysis of the local dynamics here. A brief analysis of the global dynamics is presented in Sect. 5.

## 4 Validation of the model against ERA-Interim data

In this section we compare the ERA-Interim deseasonalized jet position data with numerical simulations of our model. In order to have the same statistical sample as for the reanalysis, we simulate 37 years of daily snapshots of the jet position. The best fit of our model to the data is obtained, by a trial and error procedure, for the parameters $\eta = 1.2$, bl $= 15$, $\epsilon = 0.33$ and $\delta = 10^{-4}$. We further compare the results of model runs containing all noise terms with runs where individual terms are suppressed: coupling ($\epsilon = 0$), topography ($r = 0$) and baroclinic waves (bl $= 1$).

### 4.1 Spatiotemporal dynamics

We first consider the latitudinal distribution of the yearly median jet positions at each longitude (dots in Fig. 5) and their interannual mean (solid lines in Fig. 5). The ERA-Interim data set (Fig. 5a) presents a negative interannual mean jet position at almost all longitudes, with noticeable zonal asymmetries and a marked interannual variability. The best model run (Fig. 5b) captures both the interannual variability and, thanks to the term $r$, the longitudinal variations in average location. A run without coupling ($\epsilon = 0$) is shown in Fig. 5c. In this simulation the dynamics are local, except for the presence of block noise, resulting in a discontinuous jet profile. Unlike the ERA-Interim data, the run with no geography (Fig. 5d) has median values which are roughly symmetric around zero. Finally, the run with suppressed baroclinic activity (Fig. 5e) has a smaller interannual variability than the

ERA-Interim data and sharp changes in the median values of $x$ following the geographic constraints.

We next consider the NHJ's shifts from CJ towards NJ or SJ positions. We binarize the dynamics by the detecting all the events such that $|x| > 1$. Note that this corresponds to breaks in the jet position with the same threshold defined by BRI, although there is not exact correspondence. We then assign "0" to all the observations with $|x| < 1$ (CJ) and "1" to all the others (NJ or SJ). This procedure, known as *coding*, is widely used in dynamical systems analysis to identify different dynamical phases in complex systems (Kaneko, 1990). The so-obtained binary spatiotemporal dynamics are shown in Fig. 6a–e for all the previously described runs. In the ERA-Interim data (Fig. 6a), the switch from CJ to NJ and SJ phases occurs in clusters displaying a characteristic longitudinal extent and temporal persistence. There is also some indication of a westerly propagation of the clusters. The best model fit captures the qualitative aspects of this behavior, although the longitudinal coherence is weaker (see Sect; 4.2 below). In the remaining model simulations, the suppression of different noise terms alters either the cluster size or the westerly propagation of the clusters (Fig. 6c–e). A quantitative analysis of the cluster size spectra is presented in Fig. 6f for space clusters and Fig. 6g for time clusters. There is a clear power law behavior, reminiscent of a multiscale structure (Schertzer et al., 1997). This is coherent with the claim that the underlying jet dynamics are turbulent, with energy at all scales. Despite its simplicity, our model reproduces this power law behavior. The theoretical reasons are nontrivial and can be related to the possibility of building turbulent cascades starting from simple Langevin equations (Wouters and Lucarini, 2013; Faranda et al., 2014). We underline here the necessity of adding $\epsilon$ and having bl $> 1$. Indeed when $\epsilon = 0$, the spatial cluster spectrum consists of discrete peaks with the energy concentrated at precise scales. These are a resonance of the block noise size. When instead we impose bl $= 1$, we

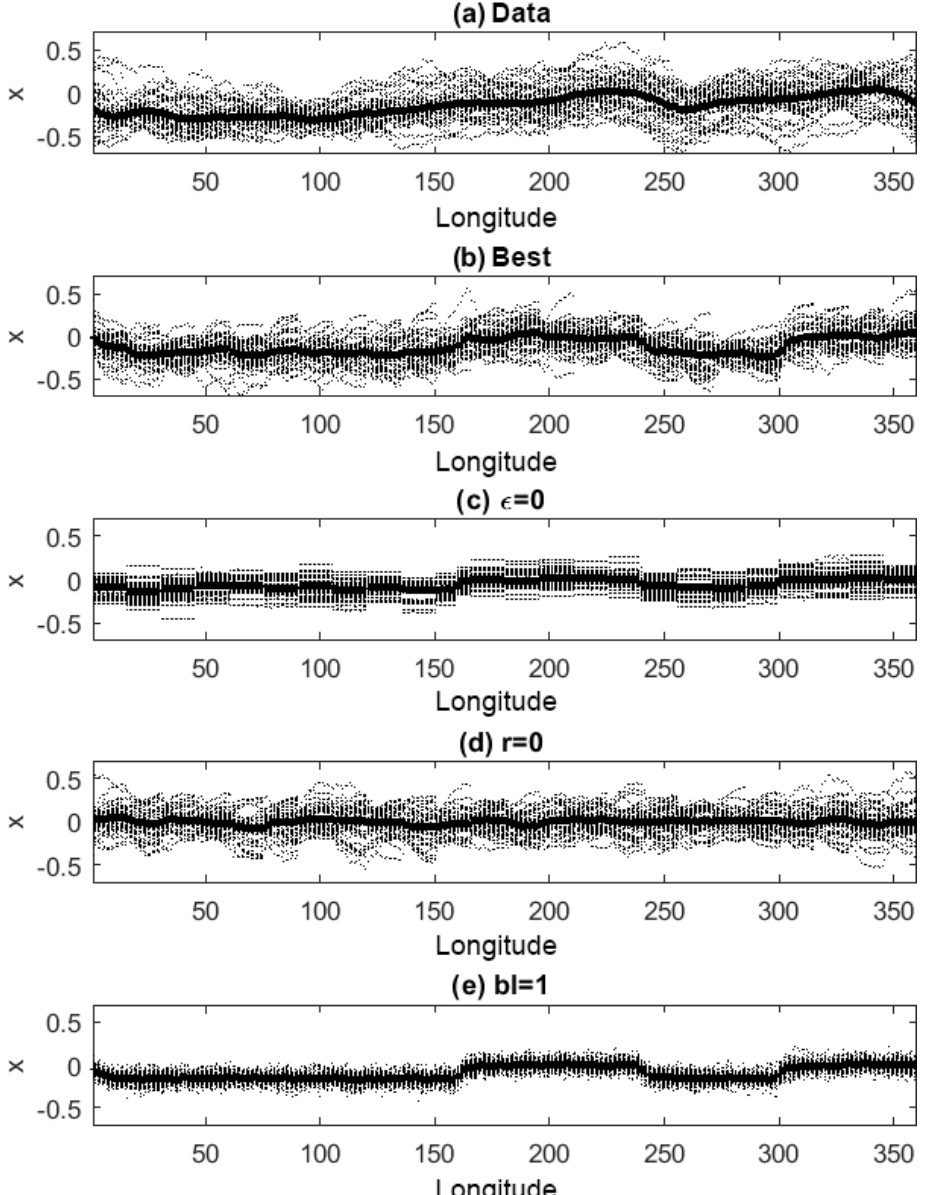

**Figure 5.** Single-year median location (dotted points) and multiyear average (solid lines) of the meridional jet position for ERA-Interim **(a)** and model **(b–e)** data. **(b)** Best-fit model, obtained with $\eta = 1.2$, bl = 15, $\epsilon = 0.33$ and $\delta = 10^{-4}$; **(c)** as in **(b)** but with $\epsilon = 0$; **(d)** as in **(b)** but with $r = 0$; **(e)** as in **(b)** but with bl = 1. The simulations consist of 37 years of daily jet positions.

still recover a power law behavior, but the slope for the temporal clustering strongly deviates from that observed in the ERA-Interim data.

## 4.2 Dynamical indicators

₅ We further assess our model by means of the $d$ and $\theta$ metrics described in Sect. 2.2, computed on both ERA-Interim data and the coupled map lattice. We also compare here the statistics of the spatial breaks of the jet, detected via the indicator BRI.

Figure 7 show the box plots of $d$ (Fig. 7a) and $\theta$ (Fig. 7b) ₁₀ for each day in the data set and BRI (Fig. 7c). The ranges of values of $d$ and $\theta$ for the ERA-Interim data resemble closely those found for sea-level pressure fields over the Northern Hemisphere (Faranda et al., 2017a). This supports the claim that the position of the jet is indicative of large- ₁₅ scale features of the NH atmospheric circulation. Similar claims about the relevance of low-dimensional projections in describing the midlatitude atmospheric circulation are presented by Madonna et al. (2017). The model runs can produce average dimensions comparable to those observed in ₂₀

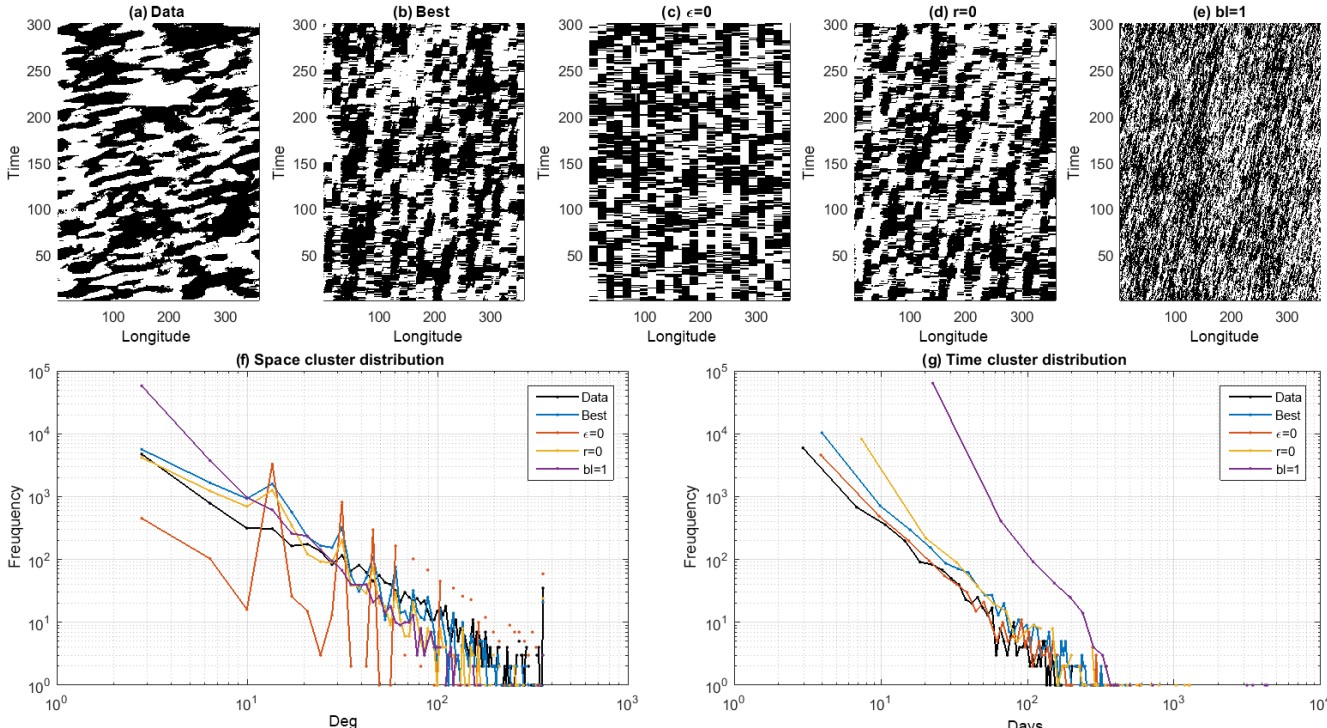

**Figure 6. (a–e)** Space–time daily representation of the binarized jet dynamics: 1 (black) corresponds to a NJ or SJ shift ($|x(t)| > 1$) and 0 (white) corresponds to a CJ position. The results are for the ERA-Interim data **(a)** and model runs **(b–e)**. The latter are the same as in Fig. 5. Space **(f)** and time **(g)** cluster spectra for the binarized ERA-interim data (black) and the different model runs (colors).

the ERA-Interim data, except for the bl = 1 case. There, the fragmented dynamics lead to a much higher dimension. This is consistent with the spatiotemporal diagrams shown in Fig. 6. The models' inverse persistence $\theta$ is slightly larger than those observed in reanalysis but still of the order of 2 d. Here we can clearly see the effect of the noise suppression ($\epsilon = 0$ and bl = 1) in modifying the dynamical properties by leading to lower persistence. Finally, we remark that the number of breaks is correlated to the local dimension. This result is consistent with Faranda et al. (2017b), who found that high $d$ matches blocking-like atmospheric configurations in the North Atlantic region. For the limiting bl = 1 case, BRI is also correlated with $\theta$: the more breaks, the lower the persistence of the flow.

## 5   Bifurcation diagram and jet regimes

The bifurcation diagram in Fig. 8 is constructed by plotting the empirical density $\rho(x)$ of the jet position at all longitudes as a function of $\epsilon$. The vertical gray line corresponds to the value of $\epsilon$ that best fits the ERA-Interim data. The diagram would look symmetric with respect to $x = 0$ if $r = 0$ everywhere, but the addition of geography via $r^{(i)}$ alters the relative proportions of time spent in SJ versus NJ. Specifically, our asymmetric land–ocean distribution implies a southward shift of the average CJ position with increasing

coupling. This is reminiscent of the behavior in the stochastic bifurcation obtained from the approximated local dynamics (Fig. 4). By analyzing the bifurcation structure of the conceptual model as a function of the coupling coefficient – which mimics the coherence of the jet – we identify three behaviors: (i) a strong and uniform jet where large meridional excursions in the jet location are relatively rare events ($\epsilon < 0.35$), which is close to the jet dynamics as inferred from the ERA-Interim data; (ii) a state with sharp meridional excursions in which the jet is very unstable and on average shifted far to the south ($0.6 < \epsilon < 0.9$); and (iii) an intermediate state of transition between the two. These jet regimes are broadly consistent with those obtained in idealized atmospheric simulations (Lachmy and Harnik, 2016; Son and Lee, 2005), although here we do not delve into the physical mechanisms underlying the different behaviors. It is also noteworthy that our model qualitatively reproduces a southern jet configuration, even though we provide it with a single NHJ and do not distinguish between eddy-driven and subtropical jets.

## 6   Conclusions

We have derived a minimal model of the jet stream position dynamics, based on a stochastic coupled map lattice, by embedding data extracted from ERA-Interim. This procedure innovates over earlier studies (e.g., Faranda et al., 2017c) by

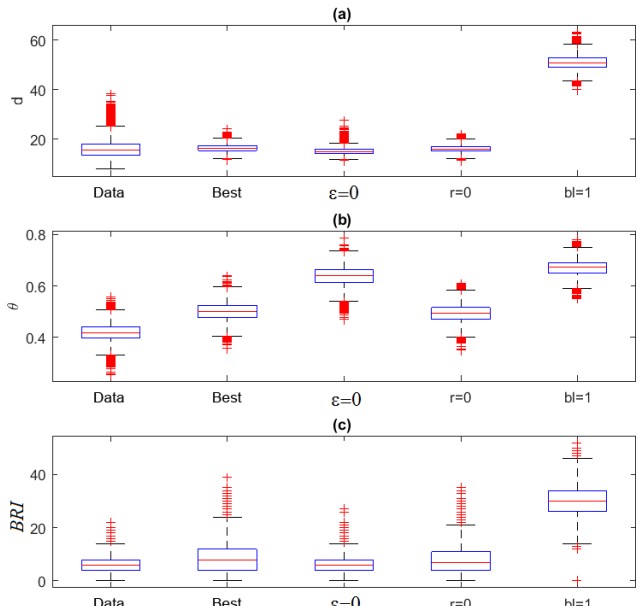

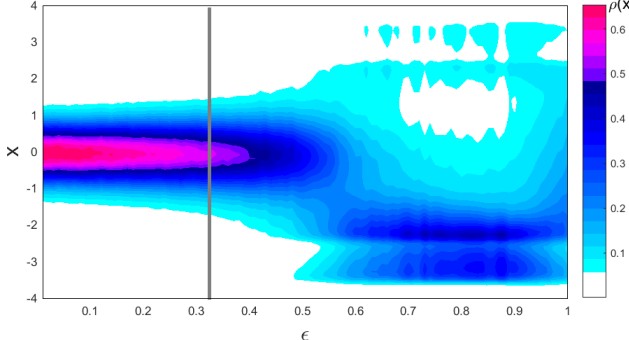

**Figure 7.** Box plots of the local dimension $d$ **(a)**, inverse persistence $\theta$ **(b)** and breaking index BRI **(c)** for the ERA-Interim data and four numerical simulations as in Fig. 5. In each box, the central mark is the median, the edges of the box are the 25th and 75th percentiles, the whiskers extend to the most extreme data points not considered outliers, and outliers are plotted individually.

**Figure 8.** Bifurcation diagram of the global dynamics obtained for $\eta = 1.2$, bl $= 15$, $0 < \epsilon < 1$ and $\delta = 10^{-4}$. The diagram represents the density of states $\rho(x)$ obtained by varying $\epsilon$. The vertical gray line indicates the value used as best fit to the ERA-Interim data.

making use of a coupled map lattice derived from a local embedding of the data and could be adapted to systems with several degrees of freedom. Instead of embedding the data of a global observable in a high-dimensional space, we have constructed the return map for the local position of the jet and then added, via coupling and noise, the physical ingredients identified in previous studies as drivers of the jet dynamics. The conceptual model is then validated and tuned using dynamical indicators of the jet's dimension and persistence in the reanalysis data.

Future analyses could apply this approach to the Southern Hemisphere, where the role of topography is less important than in its northern counterpart. This would allow us to better constrain the influence of topography on the dimension–persistence diagrams. Another possibility would be to use the low-dimensional model to build a surrogate data set of the jet positions and then apply this to atmospheric analogues, so as to construct realistic atmospheric dynamics. Finally, it would be interesting to study whether further projections of the atmospheric dynamics to a lower-dimensional space are possible, beyond the model developed here, and to test possible relations between different atmospheric blocking indices and the BRI defined here.

The analysis we have conducted can, however, already answer some of the questions left open in Faranda et al. (2017a) and Madonna et al. (2017) concerning the possibility of reducing the complex midlatitude circulation dynamics to low-dimensional representations given by blocking indices or conceptual models. The fact that the dimension–persistence diagram of our minimal model qualitatively matches many features obtained for the ERA-Interim jet position and sea-level pressure fields shows that a substantial part of the dynamics projects along a single line (the jet position). This may explain why previous investigations observed relatively low dimensions when considering the full sea-level pressure fields (Faranda et al., 2017b, a). It also suggests that breaks in the jet are responsible for higher dimensions.

**Data availability.** .TS2

Please note the remarks at the end of the manuscript.

## Appendix A: Coupled map lattice

A coupled map lattice (CML; Kaneko, 1983) is given by

$$x_{t+1}^{(i)} = (1 - \epsilon) f\left(x_t^{(i)}\right) + \frac{\epsilon}{2}\left[f\left(x_t^{(i-1)}\right) + f\left(x_t^{(i+1)}\right)\right],$$
$$(i = 1, 2, \ldots, N;\ t = 1, 2, \ldots),$$
$$\tag{A1}$$

where $\epsilon \in [0, 1]$, $x_t^{(i)} \in \mathbb{R}$ and $f(x) : \mathbb{R} \longrightarrow \mathbb{R}$. For our jet
dynamics, we adopt the open flow model, which is a class of
CML with unidirectional coupling (Kaneko, 1985):

$$x_{t+1}^{(i)} = (1 - \epsilon) f\left(x_t^{(i)}\right) + \epsilon f\left(x_t^{(i-1)}\right),$$
$$(i = 1, 2, \ldots, N;\ t = 1, 2, \ldots).$$
$$\tag{A2}$$

The CML is a phenomenological model to study complex
spatiotemporal dynamics in systems with large numbers of
degrees of freedom. The idea is to discretize the dynamics
in space and time, while capturing the global phenomenol-
ogy of the system. CMLs have been successfully applied to
processes such as turbulence in thermal convection (Yanagita
and Kaneko, 1993) and turbulent puff-in-pipe flow (Avila and
Hof, 2013). It is convenient for us to model the jet dynamics
leveraging the CML approach because we can extract a local
one-dimensional map from the observed time series.

## Appendix B: Average return map and noise

To extract the local jet dynamics, we construct an average
return map. We first coarse-grain the state space into $M$ par-
titions $L_k^{(i)}$ ($k = 1, 2, \ldots, M$) and let $\overline{x}^{(i,k)}$ be the midpoint
of $L_k^{(i)}$. Then, we construct a set $Y^{(i,k)} = \{x_t^{(i)} | x_{t-1}^{(i)} \in L_k^{(i)}\}$
($t = 2, 3, \ldots$) and a return map $f^{(i)}$ via the return plot of
$(\overline{x}^{(i,k)}, \langle Y^{(i,k)}\rangle)$, where $\langle \cdot \rangle$ is the average over the elements
of $Y^{(i,k)}$ at each longitude $i$ and at each partition $k$:

$$\langle Y^{(i,k)}\rangle = f^{(i)}\left(\overline{x}^{(i,k)}\right),\ i = 1, 2, \ldots, N,\ k = 1, 2, \ldots, M,\ \ (\text{B1})$$

where $|x| \leq c$. In the region $|x| > c$, we assume linear re-
flection effects. As a result, we have the return map $f^{(i)}$ in
Eq. (3). Figure 2 illustrates the construction for $i = 1$ and
$M = 500$.

An important ingredient of the jet dynamics is the pres-
ence of topographic obstacles to the midlatitude zonal flow.
Mountain ranges and land–sea boundaries cause meridional
deviations in the mean jet location (Tibaldi et al., 1980).
This inhomogeneity can be modeled via a parameter $r^{(i)}$
that mimics this "spatial noise". Since the topography is at
most a few kilometers in height, this translates to a per-
turbation of the order of $10^{-3}$ in the model. Reasonable
geographical constraints are therefore $r^{(i)} = -0.02 (i \in \text{land})$
and $r^{(i)} = 0.0 (i \in \text{ocean})$, where "land" spans the ranges $0 \leq$
$i < 161$ and $239 \leq i < 301$ and "ocean" spans the ranges
$161 \leq i < 239$ and $301 \leq i < 360$. The negative sign for the
jet shifts over land is justified by the negative median values
of the ERA-Interim jet position anomalies over land (com-
pare Fig. 5a and b with Fig. 5c where no topography is
present).

As discussed in Sect. 3.2, noise is a fundamental ingre-
dient in the jet dynamics. The "turbulent noise" term $\nu$ re-
lates to physical phenomena in the range of a few meters
to a few kilometers, implying a perturbation in the range
$10^{-4} < \nu < 10^{-3}$, where $\nu$ is a random variable obeying the
uniform distribution. The second noise term, $\eta$, relates to
baroclinic activity, and we model it as a block noise taking
the same value over bl blocks (the one-dimensionalized size
of cyclones or anticyclones in our model) with an amplitude
of the order of 1. The latter value is determined empirically
as it is an indicative magnitude of the large shifts midlati-
tude baroclinic systems can induce in the jet. To determine
a realistic length for bl, we reason as follows: given that our
model has a reference scale of about 100 km, and assuming a
typical scale for extratropical cyclones of about 3000 km, we
then have that bl $\approx$ 30 blocks. However, the perturbations are
associated with the cyclone radius rather than diameter: up-
stream of the cyclone, the jet will mostly be deviated south-
wards, while downstream of the cyclone, the jet will mostly
be deviated northwards. We therefore take the block pertur-
bation to be of the size bl = 15 blocks.

Owing to the unidirectional coupling in our lattice
jet model and to the large $N$, the local dynamics can
be approximated by a nonautonomous dynamical system
$x_{t+1}^{(i)} \simeq f^{(i)}(x_t^{(i)}) + p_t^{(i)}$, where the external force $p_t^{(i)} =$
$\epsilon[f^{(i-1)}(x_t^{(i-1)}) - f^{(i)}(x_t^{(i)})] + \nu_t^{(i)} + \eta_t^{(i)}$. Assuming that the
time averages $\langle f^{(i-1)}(x_t^{(i-1)}) - f^{(i)}(x_t^{(i)})\rangle$, $\langle \nu_t^{(i)}\rangle$ and $\langle \eta_t^{(i)}\rangle$ are
all 0 by symmetry, we have $\langle p_t^{(i)}\rangle \approx 0$. Thus, we recover the
average return map given in Eq. (B1).

**Supplement.** The supplement related to this article is available online at: https://doi.org/10.5194/esd-10-1-2019-supplement.

**Author contributions.** DF and YS performed the analysis and derived the conceptual model. GM computed the jet position data. All the authors participated in the writing and the discussions. CE3

**Competing interests.** The authors declare that they have no conflict of interest. TS3

**Acknowledgements.** All the authors were supported by the ERC grant no. 338965-A2C2. Davide Faranda and Yuzuru Sato were supported by the CNRS PICS grant no. 74774 and by London Mathematical Laboratory External Fellowships. Davide Faranda and Pascal Yiou were supported by an INSU-CNRS-LEFE-MANU grant (project DINCLIC). Yuzuru Sato was supported by the Grant in Aid for Scientific Research (C) no. 18K03441, JSPS, Japan. Gabriele Messori was supported by grant no. 2016-03724 from the Swedish Research Council Vetenskapsrädet TS4 and the Department of Meteorology of Stockholm University. Yuzuru Sato, Gabriele Messori and Nicholas R. Moloney thank the LSCE for hospitality.

**Financial support.** This research has been supported by the H2020 European Research Council (grant no. A2C2 (338965)), the Centre National de la Recherche Scientifique (grant no. PICS 74774) and the JSPS (grant no. 18K03441). TS5

**Review statement.** This paper was edited by Andrey Gritsun and reviewed by two anonymous referees.

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

## Remarks from the language copy-editor

CE1    Please note that Fig. 3 was edited during copy-editing. Please review the figure content carefully.

CE2    Please define.

CE3    Please review the content of the following sections carefully, as edits are not displayed in the track-changes PDF: "Financial support", "Author contributions" and "Competing interests".

## Remarks from the typesetter

TS1    The composition of Figs. 1, 3 and 5–7 has been adjusted to our standards.

TS2    Please provide a statement on how your underlying research data can be accessed. If the data are not publicly accessible, a detailed explanation of why this is the case is required. The best way to provide access to data is by depositing them (as well as related metadata) in reliable public data repositories, assigning digital object identifiers (DOIs), and properly citing data sets as individual contributions. Please indicate if different data sets are deposited in different repositories or if data from a third party were used. Additionally, please provide a reference list entry including creators, title, and date of last access. If no DOI is available, assets can be linked through persistent URLs to the data set itself (not to the repositories' home page). This is not seen as best practice and the persistence of the URL must be secured.

TS3    This sentence has been adjusted to our standards. Please confirm.

TS4    Please check word here.

TS5    Please note that the funding information has been added to this paper. Please check if it is correct. Please also double-check your acknowledgements to see whether repeated information can be removed or changed accordingly. Thanks.

TS6    Please provide page range or article number and DOI number.

TS7    Please provide DOI number.

TS8    Please provide place of publication.

TS9    Please provide all author names.

TS10    Please provide page range or article number and DOI number.

TS11    Please provide page range or article number and DOI number.

TS12    Please provide DOI number.

TS13    Please provide DOI number.

TS14    Please provide DOI number.

TS15    Please provide DOI number.

TS16    Please provide place of publication.

TS17    Please provide full page range.

TS18    Please provide full page range.

TS19    Please provide date and place of seminar.

TS20    Please provide DOI number.

TS21    Please provide volume and page range or article number and DOI number.

TS22    Please provide full page range.

TS23    Please provide volume.

TS24    Please provide page range or article number and DOI number.

TS25    Please provide year of publication.

TS26    Please provide full page range.

TS27    Please provide journal name, volume and page range (or article number and DOI number) or publisher and place of publication.

TS28    Please provide full page range.

TS29    Please provide page range or article number and DOI number.