# Peer review of "Minimal dynamical systems model of the northern hemisphere jet stream via embedding of climate data"

_Earth System Dynamics, 2018_

## Referee Comment (RC1) · Anonymous Referee #1 · 22 Jan 2019

Review

"Minimal dynamical systems model of the northern hemisphere jet stream via embedding climate data"

Authors: D Faranda, Y Sato, G Messori, NR Moloney & P Yiou

Review of esd-2018-80

Recommendation: Major revision.

Summary:

[Figure]

The authors develop a simple stochastic toy model of the latitudinal position of the peak northern hemisphere upper tropospheric jet stream at the different longitudes. Their aim is to study the jet variability due to transitions from wave breaking and block formation. The simple model uses atmospheric data to determine the average functional form f Ì̆E(x) of the processes that determine the current location and the effects of small scale subgrid processes such as convection and gravity waves, topographic processes, and baroclinic Rossby wave processes are represented by three stochastic terms. The stochastic subgrid terms are tuned to represent some of the broad statistical properties of observations of the latitudinal representation of the peak northern hemisphere jet.

General Comments:

The average functional form f Ì̆E(x) and the three stochastic subgrid terms are purely heuristic so unlike the case of other reduction techniques and subgrid modeling the connection with the physics of the problem is unclear. How would the results change with different, perhaps more physical, subgrid terms? Without a physical basis for the driving terms it seems unlikely that the simple model will be seen as any more than a curve fitting exercise.

The presentation of the article is substandard and not in a form that would appeal to the audience of ESD. The paper lacks motivation, the mathematics is poorly presented with terms undefined and too many typos and has the feel of a first draft. Perhaps unfortunately, the mathematical nomenclature for what are really very simple concepts (new words for old), would most likely put off an audience of largely data analysts. For this audience the authors should make the article more pedagogical and stand alone.

Specific Comments:

P2, line8: Perhaps references to Charney and De Vore (1979) and Wiin-Nielsen (1979) would be appropriate. Section 3: The mathematics is surprisingly poorly presented given that one of the authors is from a Department of Mathematics and Statistics. For example, you need to define n as the time step, i as the longitude and define N=360

when it first appears. You need to check your equations for typos as in equation (3). Also, the equations keep changing until you eventually settle on the system that you eventually address. P4, lines 2&3: Northern hemisphere blocking occurs in preferred regions so why does the return map not reflect that? P4, lines 4-31: Why is necessary to have separate stochastic processes for the effects of (1) convection and gravity waves, (2) effects of topography and (3) effects of baroclinic Rossby waves, rather than combine the three? Also why are these parameterizations purely stochastic when more systematic subgrid parameterizations indicate that they should be represented by a combination of deterministic and stochastic terms (e.g., Kisios and Frederiksen 2018 and references therein). In general, the authors should relate their subgrid parameterizations at least in broad terms to physically based parameterizations. P4, lines 14-19: The impression that the authors convey here is that the topography is a stochastic term in their model in which case it should be multiplicative noise rather than additive noise. However, according to the above reference deterministic topography interacting with eddies produces an additive noise contribution as well as contributions from barotropic and baroclinic Rossby waves. P4, line20: baroclinic –> baroclinic and barotropic P4, line 21: 10ˆ-3 –> 10ˆ3 P5, line 5: What exactly is the form of the non-autonomous force? What is the explicit time dependence? You should define your terms for an audience of largely data analysts. Section 4: Again the mathematics is poorly presented. I would expect precision and elegance from mathematicians. You will need to explain your terminology for the major audience of ESD.

The authors need to carefully check their manuscript for a number of typos.

References: Kitsios, V., and J. Frederiksen, 2018: Subgrid parameterizations of the eddy-eddy, eddy-meanfield, eddy-topographic, meanfield-meanfield and meanfield-topographic interactions in atmospheric models. J. Atmos. Sci. doi:10.1175/JAS-D-18-0255.1

Please also note the supplement to this comment:

https://www.earth-syst-dynam-discuss.net/esd-2018-80/esd-2018-80-RC1-supplement.pdf

---

## Referee Comment (RC2) · Anonymous Referee #2 · 22 Jan 2019

This paper proposes a stochastic coupled map lattice (CML) model to describe dynamics of latitudinal position of the Northern Hemisphere atmospheric jet at each longitude with a stated goal to evaluate how this model represents the dynamical features of the jet. The manuscript needs to be substantially improved before I can recommend it for publication. In particular, presentation of the CML model lacks clarity for general readership, as well as interpretation and significance of some results are overstated.

Comments:

1. Please provide some background on CML and why it has been chosen for this study.

2. Please provide more mathematical details on return map in Section 3 and how it

can be used to estimate f(x).

3. Why the particular form of Eq (4) is chosen and how these coefficients are esti-
mated?

4. What about uncertainties in the model coefficients? Fig.3 shows that red line (Eq.4)
seem to be missing excursions that are very few to begin with.

5. It is rather hard to follow the discussion of the stochastic noise terms and it leaves
impression that they are tuned without much mathematical guidance.

6. The Fig.7 comparison of summary statistics (ACF and PDF) for the optimal value
of epsilon = 0.4 does not show much qualitative agreement between the modeled and
observed dynamics (also in P15 in conclusions). The space-time patterns also look
visibly rather different. It makes look weaker the rest of results on bifurcation analysis
and dynamical indicators.

---

## Author Response (AR1)

**Answer Letter: Minimal dynamical systems model of the northern hemisphere jet stream via embedding of climate data by Faranda et al.**

esd-2018-80

Dear Editor,

We are pleased to resubmit a new version of our paper "Minimal dynamical systems model of the northern hemisphere jet stream via embedding of climate data" for consideration in ESD. We have undertaken the extensive changes recommended by the reviewers and yourself and rewritten/extended the paper to make it more readable to the climate scientists community. The specific changes are given in the answers below, however we would like to underline the major changes in this manuscript version:
- The introduction has been revised: a better overview of the existing literature on the jet dynamics is provided as well as a motivation for the use of low dimensional models. The added value of stochastic modelling is also underlined
- The methods section has been rewritten to be more readable
- The model section has been rewritten and each single term is motivated on physical basis and linked to the existing literature, as recommended by the reviewers and yourself.
- The validation of the model has been rewritten: we have looked in the phase space for the best parameters and presented indicators that are more readable for the climate community.
- Following the request to provide a rigorous introduction on coupled map lattices and the embedding procedure, we have added two appendices.

At the end of the referees answers we also provide a marked-up version of the manuscript, useful for review purpose. We hope that this new version of the article will be suitable for publication in ESD.

Best Regards,
Davide Faranda,
On behalf of the authors

**Referee 1**

General Comments:
The average functional form f(x) and the three stochastic subgrid terms are purely heuristic so unlike the case of other reduction techniques and subgrid modeling the connection with the physics of the problem is unclear. How would the results change with different, perhaps more physical, subgrid terms? Without a physical basis for the driving terms it seems unlikely that the simple model will be seen as any more than a curve fitting exercise.
The presentation of the article is substandard and not in a form that would appeal to the audience of ESD. The paper lacks motivation , the mathematics is poorly presented with terms undefined and too

many typos and has the feel of a first draft. Perhaps unfortunately, the mathematical nomenclature for what are really very simple concepts (new words for old), would most likely put off an audience of largely data analysts. For this audience the authors should make the article more pedagogical and stand alone.

**We thank the reviewer for this comment and we take into serious consideration the criticism that our paper should be better motivated. We disagree with the statement that the embedding methodology we apply amounts to curve-fitting (indeed, the only part of our analysis where this definition may be argued for is for obtaining the return map). This model is motivated by geometrical and temporal evolutions of the jet (in a reanalysis). Virtually any models, including conventional numerical simulations of large-scale atmospheric flows, require some arbitrarily chosen parameters, and our case is no different. We also underline that our coupled map lattice model rests on clear physical hypotheses, such as: 1) the eastward propagation of information within the jet stream, 2) the presence of anticyclones and cyclones (baroclinic activity) i.e. sinuosity of the jet, 3) the presence of geographical constraints, 4) small-scale turbulent disturbances. This again sets it apart from curve-fitting exercises. We indeed view our approach as complementary to other idealised approaches which have attempted to formalise atmospheric waves and sinuosity, such as that of Petoukhov et al. In contrast to the latter, our model does not rely on regular wave decomposition hypotheses that are difficult to verify in practice. We now address this aspect more directly in the introduction to clarify the motivations underlying our analysis.**

**However, we do agree with the Reviewer that our illustration of the physical principles underlying some of our choices were not as clear as they should have been. We now detail how our choices have solid physical underpinnings, issued from both laboratory tank experiments, numerical simulations in the literature and scale arguments applied to fundamental concepts in atmospheric dynamics. Concerning the physical processes specific to sub-grid scales, we highlight that a key advantage of the return plot methodology is that it enables us to ignore detailed microphysics, focusing instead on the largest scale effective dynamics which is observed in the real data. Indeed, such small-scale processes are only indirectly present in the data we use to build our model, through assimilation of observations, but would be largely parametrized in the numerical model underlying the reanalysis dataset. Finally, we would like to stress that our purpose is to develop a minimal model to study the phenomenology of the effective dynamics of the jet flow, emerging from the complex underlying physics. Such a model does not require physical sub-grid terms a priori, but only if they were found to be essential to capture the large-scale phenomenology – which we show is not the case. This approach is fundamentally different from Direct Numerical Simulation (DNS) based studies, which typically start from the detailed microphysics at the cost of not incorporating real large-scale data. The phenomenological properties of our model, such as its bifurcation structure, are largely independent of the selected parameters, except for a few leading terms such as kappa, beta, and epsilon. We thus believe that our approach is a valid complement to the classical DNS-like approaches.**

**Concerning the second part of the Reviewer's comment, we address this in more detail in the responses to the specific comments below. One point we would like to highlight is that we have taken very seriously the Reviewer's encouragement to refocus our article to appeal to ESD readership, especially when explaining the derivation and implementation of the model. To this effect, amongst**

other changes we have added two appendices providing background on some key concepts leveraged in the paper: Appendix A: Coupled map lattice, and Appendix B: Average return map and noise.

Specific Comments:
P2, line8: Perhaps references to Charney and De Vore (1979) and Wiin-Nielsen (1979) would be appropriate.

**We have added these references, as suggested.**

Section 3: The mathematics is surprisingly poorly presented given that one of the authors is from a Department of Mathematics and Statistics. For example, you need to define n as the time step, i as the longitude and define N=360 when it first appears. You need to check your equations for typos as in equation (3). Also, the equations keep changing until you eventually settle on the system that you eventually address.

**We will replace n with t, and now describe all variables and the full model at the beginning of Section 3. We will further include a brief background review on coupled map lattices (CMLs), to highlight why they are appropriate in our context, in addition to providing a more detailed background in Appendix A.**

P4, lines 2&3: Northern hemisphere blocking occurs in preferred regions so why does the return map not reflect that?

**The local topography is represented as small shifts in the return maps. These small differences in the maps induce a sudden large change of the dynamics, corresponding to shifting the jet towards the north or the south and therefore triggering blocking in selected regions. We omit detailed landscape factors such as high mountains in this particular model for studying global phenomenology. However, these may be included via the boundary condition $r^{(i)}$. We now show that the inclusion of $r^{(i)}$ is able to highlight preferred regions where the jet shifts towards northern or southern directions (Figure 1 of this answer). We further argue that it is a strength of the model that it reproduces jet-like phenomenology, independently of the choice of the location, and that geographic effects can be introduced via boundary conditions. This discussion will be added to the new version of the paper.**

P4, lines 4-31: Why is necessary to have separate stochastic processes for the effects of (1) convection and gravity waves, (2) effects of topography and (3) effects of baroclinic Rossby waves, rather than combine the three?

**Given the CML model, the external perturbations to each local dynamics are categorized as: (0) initial conditions (1) local noise, (2) spatial boundary conditions, and (3) global noise. In this study, we made models for factors (1), (2), and (3). Term (0), the initial conditions, is chosen from a stationary state in the model. These are based mainly on (1) effects of convection and gravity waves, (2) effects of topography, and (3) effects of baroclinic Rossby waves. In brief, we started with modeling**

**phenomenological external perturbations, and then verified the underlying physics which affects factors (1), (2), and (3). To answer the question of the Reviewer, we cannot combine the three terms into one because they act on different spatial scales. More specifically: without (1) the system will be stacked in only one of the three states with no transitions; without (2) the jet position will not have a geographical dependence (see again Figure 1) and would thus not match the patterns observed in the ERA-Interim data; without (3) there will not be persistent blocking. We will add to the paper the new Figure 2, which shows temporal and spatial cluster size distribution for different models and the data, once they have binarized as follows: "1" means a northern shift of the jet with respect to its central position, "0" means a southern shift. The different model runs show the effect of the suppression of noise terms. The figure clearly shows that by suppressing one of the noise ingredients, the spatiotemporal cluster distributions of the data cannot be reproduced. The motivation of our work is precisely to show that these three ingredients are essential to reproduce the features of the jet dynamics.**

Also why are these parameterizations purely stochastic when more systematic subgrid parameterizations indicate that they should be represented by a combination of deterministic and stochastic terms (e.g., Kisios and Frederiksen 2018 and references therein). In general, the authors should relate their subgrid parameterizations at least in broad terms to physically based parameterizations.

**We will add a discussion about the stochastic vs deterministic parametrization based on Kisios and Frederiksen 2018 and references therein. Our goal here is to have only the large scales described by a deterministic term, as we build a global model of the jet dynamics. Indeed, a mixture of deterministic and stochastic terms improve the dynamical description of the jet dynamics, but the addition of other terms will not make our model a minimal model of the jet dynamics.**

P4, lines 14-19:
The impression that the authors convey here is that the topography is a stochastic term in their model in which case it should be multiplicative noise rather than additive noise. However, according to the above reference, deterministic topography interacting with eddies produces an additive noise contribution as well as contributions from barotropic and baroclinic Rossby waves.

**In our model, the topography is given as a boundary condition and therefore is a deterministic term. We will rephrase the model description to say that it is included in the perturbations term, where the perturbations are split into deterministic (topography) and stochastic (turbulence and baroclinic waves) contributions.**

P4, line20:
baroclinic –> baroclinic and barotropic

**Corrected.**

P4,line 21:

10ˆ-3 –> 10ˆ3

**Corrected.**

P5, line 5:
What exactly is the form of the non-autonomous force? What is the explicit time dependence? You should define your terms for an audience of largely data analysts.

**The model of the local dynamics at location i can be rewritten as:**
**$x_{t+1}^{(i)}= f(x_t^{(i)})+p_t^{(i)}$.**
**The non-autonomous term $p_t^{(i)}$ includes all driving forces other than $f(x_t^{(i)})$ at position i, and its explicit time dependence cannot be given simply. Non-autonomous dynamical systems theory can be applied to dynamical systems with such "unknown" driving forces. Here, we approximated it as a random variable $p_t^{(i)}$ in [-kappa, kappa], assuming a bounded external force, and analyzed the bifurcation structure of the approximated model. We clearly explain the above mathematical approach in the revised Section 3 in the manuscript.**

Section 4:
Again the mathematics is poorly presented. I would expect precision and elegance from mathematicians. You will need to explain your terminology for the major audience of ESD. The authors need to carefully check their manuscript for a number of typos.

**We will add the mathematical details on the dynamical indicators in two appendices mentioned in the reply to the Reviewer's general feedback.**

References: Kitsios, V., and J. Frederiksen, 2018: Subgrid parameterizations of the eddy-eddy, eddy-meanfield, eddy-topographic, mean field-mean field and mean field topographic interactions in atmospheric models. J. Atmos. Sci. doi:10.1175/JAS-D-18-0255.

**Referee 2**

The manuscript needs to be substantially improved before I can recommend it for publication. In particular, presentation of the CML model lacks clarity for general readership, as well as interpretation and significance of some results are overstated.

**Both Reviewers have highlighted lack of clarity for ESD readership as a key shortcoming of our manuscript, and we have taken this comment very seriously. In addition to the changes detailed in the**

**replies to the individual comments below, we have added two appendices providing background on some key concepts leveraged in the paper: Appendix A: Coupled map lattice, and Appendix B: Average return map and noise.**

Comments:

1. Please provide some background on CML and why it has been chosen for this study.

**We will describe the full model at the beginning of Section 3, and further include a brief background review of CMLs, motivating their use here, as an appendix in the new version of the manuscript.**

2. Please provide more mathematical details on return map in Section 3 and how it can be used to estimate f(x).

**We have added a clearer mathematical background on return maps in Section 3, and also highlight its advantages in the present context.**

3. Why the particular form of Eq (4) is chosen and how these coefficients are estimated?

**The particular form of equation 4 is chosen as the one best fitting the data and presenting a stable state around 0. We have tried other functional forms for the maps given in Eq. 4. An account of this will be given in Section 3 in the revised version of the study.**

4. What about uncertainties in the model coefficients? Fig. 3 shows that red line (Eq.4) seem to be missing excursions that are very few to begin with.

**The return map is obtained by adapting the model to the data. The phenomenological properties, such as bifurcation structure, are largely independent of the selected parameters. The excursions are modelled via the stochastic escape that will add fluctuations on top of the red line. We will explain this in the new version of the manuscript.**

5. It is rather hard to follow the discussion of the stochastic noise terms and it leaves impression that they are tuned without much mathematical guidance.

**In the new version we will explain the rationale behind: (1) local noise, (2) spatial boundary conditions, and (3) global noise, more clearly in the manuscript. We have further made an effort to highlight that these issue from both mathematical and physical considerations, which ground our model in both dynamical systems theory and atmospheric dynamics.**

6. The Fig.7 comparison of summary statistics (ACF and PDF) for the optimal value of epsilon = 0.4 does not show much qualitative agreement between the modeled and observed dynamics (also in P15 in conclusions). The space-time patterns also look visibly rather different. It makes look weaker the rest of results on bifurcation analysis and dynamical indicators.

**Indeed, we realized how important it is to provide a quantitative characterization of the spatio-temporal properties of the model versus data. We proceed as follows: we binarize the data so that "1" is any shift towards the northern jet state and "0" is a shift towards the southern jet state. We then compute the time and spatial cluster size distributions for different models, including or not the noise terms and show their importance in matching the distribution of the jet position observed in the data. We will replace Figure 7 in the paper with the new Figure 2 here in which we show these analyses. The best model is now obtained for the parameters eta=1.2, epsilon=0.33, r_oceans=0, r_mountains=0.02,**

[Figure]

Figure 1: Role of the term r_i in the shift of the jet position towards northern or southern latitudes. The figures show the fraction of shifts towards the north: a value >0.5 indicates that the jet's preferred position is to the north, a value < 0.5 that the jet's preferred position is to the south. a) Model with r_i=0 over oceans and r_i=-0.02 over the mountains (same domains as given in the previous version of the paper). b) Model with r_i=0 for all the latitudes. Red: shift frequency from data. Black: shift frequency from the model: each line corresponds to a realization of the system.

[Figure]

Figure 2: Upper plots: Temporal and spatial cluster size distribution for the ERA Interim data (top left), and few different model runs. The clusters are obtained once the data are binarized: 1 corresponds to a northern shift of the jet with respect to its central position; 0 corresponds to a southern shift. The different model runs show the effect of the suppression of noise terms. Lower plots: Space and time cluster distributions for ERA interim data (black) and different model runs (colors).

0394Δ 8710Δ 0331 014B

[revised manuscript text omitted]

**Figure 7.**  Boxplots of the local dimension $\cancel{D}$ $d$ (a,c,e), inverse persistence $\cancel{\Theta}$ $\theta$ (b,) and breaking index $BRI$ (c)for  the  $\cancel{\delta = 10^{-3}, r^{(i)} = 0.02(i \in \text{land})}$ ERA Interim data and $\cancel{r^{(i)} = -0.04(i \in \text{ocean}), bl = 15}$four numerical simulations as in Figure 5. In each box,  $\cancel{\delta = 10^{-4}}$the central mark is the median, $\cancel{r^{(i)} = 0.01(i \in \text{land})}$ the edges of the box are the 25th and $\cancel{r^{(i)} = -0.02(i \in \text{ocean})}$75th percentiles, $\cancel{bl = 15. \text{ e}}$the whiskers extend to the most extreme data points not considered outliers, $\cancel{f) \delta = 10^{-4}, r^{(i)} = 0.01(i \in \text{land})}$ and $\cancel{r^{(i)} = -0.02(i \in \text{ocean}), bl = 20}$outliers are plotted individually.

[Figure]

**Figure 8.** Bifurcation diagram of the global dynamics obtained for $\eta = 1.2$, $bl = 15$, $0 < \epsilon < 1$, and $\delta = 10^{-4}$. The diagram represents the density of states $\rho(x)$ obtained by varying $\epsilon$. The vertical grey line indicates the value used as best fit to the ERA Interim data.

---

## Referee Report (RR1)

**Review**

"Minimal dynamical systems model of the northern hemisphere jet stream via embedding climate data"

Authors:  D Faranda, Y Sato, G Messori, NR Moloney & P Yiou

Review of esd-2018-80 version 2

Recommendation: Accept after very minor corrections.

General Comments:

The article is now very much improved and generally suitable for publication with just a few typos and cosmetic changes to be fixed. It provides analysis techniques which should be of general interest to the ESD readership as well as a very simple data driven model of regime transitions of the atmospheric zonal flow.

Specific Comments:

P6, line 30: 2018 → 2019

P8, line 23: "There is also some indication of westerly propagation of the clusters": I found this a bit puzzling at first since blocks generally develop upstream of the blocking region and then lock into place with some subsequent oscillations about the central position. However, the largest signal will probably come from the largest scale planetary waves which will tend to retrogress. So maybe the retrogression is not a signal of blocking so much as of planetary wave retrogression? No action is required here.

P8, lines 29-30: Brackets around the references.

P9, line 4: ?? → 7

P9, line 12: sea → see

References:

The references need to all be made consistent with ESD requirements in terms of page numbers, capitals and shortening of journal titles etc. The following reference was incomplete:

Kitsios, V., and J. Frederiksen, 2019: Subgrid parameterizations of the eddy-eddy, eddy-meanfield, eddy-topographic, meanfield-meanfield and meanfield-topographic interactions in atmospheric models. J. Atmos. Sci. **76**, 457-477, (2019). doi:10.1175/JAS-D-18-0255.1

---

## Author Response (AR2)

Dear Editor,

We have revised the paper "Minimal dynamical systems model of the northern hemisphere jet stream via embedding of climate data" according to the reviewers' and editorial suggestions. A detailed overview of the answers is given below. Thank you for considering our work for publication in Earth System Dynamics.

Best Regards,

Davide Faranda

```
REFEREE 1

Recommendation: Accept after very minor corrections.

General Comments:

The article is now very much improved and generally suitable for
publication with just a few typos and cosmetic changes to be fixed. It
provides analysis techniques which should be of general interest to
the ESD readership as well as a very simple data driven model of regime
transitions of the atmospheric zonal flow.
```

We thank the reviewer for appreciating our revision.

```
Specific Comments:

P6, line 30: 2018 -> 2019
```

Corrected

```
P8, line 23: "There is also some indication of westerly propagation of
the clusters": I found this a bit puzzling at first since blocks
generally develop upstream of the blocking region and then lock into
place with some subsequent oscillations about the central position.
However, the largest signal will probably come from the largest scale
planetary waves which will tend to retrogress. So maybe the
retrogression is not a signal of blocking so much as of planetary wave
retrogression? No action is required here.
```

This is definitely an interesting line of research. In a future work, we will address this problem by extending the current model with the information ofn the maximum velocity

```
P8, lines 29-30: Brackets around the references.

P9, line 4: ?? -> 7

P9, line 12: sea > see
```

corrected

```
References:
```

The references need to all be made consistent with ESD requirements in
terms of page numbers, capitals and shortening of journal titles etc.
The following reference was incomplete
Kitsios, V., and J. Frederiksen, 2019: Subgrid parameterizations of the
eddy-eddy,
eddy-meanfield, eddy-topographic, meanfield-meanfield and meanfield-
topographic
interactions in atmospheric models. J. Atmos. Sci. 76, 457-477, (2019).
doi:10.1175/JAS-D-18-0255.1

corrected

REFEREE 2

Authors have substantially improved the paper after revision. Most of
the critical comments from previous round have been also addressed.
Still, it is not clear how exactly the optimal model parameters were
estimated, for example A, beta, c in equation 3, and parameters of the
noise (line 4 on p. 8). Was it by trial and error or some cost function
was minimized? Either way, it should be clarified in the text.

We thank the referee for the comments. We have specified tha we used a trial and error procedure
(page 8 line 4)